# The Experiences of Stakeholders Using Social Media as a Tool for Health Service Design and Quality Improvement: A Scoping Review

**DOI:** 10.3390/ijerph192214851

**Published:** 2022-11-11

**Authors:** Louisa Walsh, Nerida Hyett, Nicole Juniper, Chi Li, Sophie Hill

**Affiliations:** 1Centre for Health Communication and Participation, School of Psychology and Public Health, La Trobe University, Melbourne, VIC 3083, Australia; 2La Trobe Rural Health School, La Trobe University, Bendigo, VIC 3550, Australia; 3Orygen, Parkville, VIC 3052, Australia; 4Albury Wodonga Health, Wodonga, VIC 3690, Australia

**Keywords:** social media, quality improvement, health services, system transformation, digital health, stakeholder engagement, consumer engagement, health policy

## Abstract

Background: Health organisations and stakeholders use social media for a range of functions, including engaging stakeholders in the design and quality improvement (QI) of services. Social media may help overcome some of the limitations of traditional stakeholder engagement methods. This scoping review explores the benefits, risks, barriers and enablers for using social media as a tool for stakeholder engagement in health service design and QI. Methods: The searches were conducted on 16 August 2022. Inclusion criteria were: studies of any health service stakeholders, in any health setting, where social media was used as a tool for service design or QI. Data was analysed using deductive content analysis. A committee of stakeholders provided input on research questions, data analysis and key findings. Results: 61 studies were included. Benefits included improved organisational communication and relationship building. Risks/limitations included low quality of engagement and harms to users. Limited access and familiarity with social media were frequently reported barriers. Making discussions safe and facilitating access were common enablers. Conclusion: The benefits, risks, barriers and enablers identified highlight the complexity of social media as an engagement tool for health service design and QI. Understanding these experiences may help implementers design more effective social media-based engagement activities.

## 1. Introduction

Social media has shaped the internet since Web 2.0 was defined in 2004 [1]. Social media are any online platforms that allows users to create or curate content, interact and form social networks [2]. Its use is extremely widespread, with over 4.2 billion people worldwide having at least one social media account [3]. People use social media sites for a variety of health-related purposes, including searching for health information [4,5,6,7], finding support for their condition [5,7,8] and as an aid for self-management [7,8]. Health organisations also use social media as way to increase consumer access to health information [6].

Social media can also be used to facilitate the engagement consumers and other stakeholders in activities to change, design, or improve health services [9]. Batalden and Davidoff (2007) [10] define quality improvement (QI) in health as “the combined and unceasing efforts of everyone—healthcare professionals, patients and their families, researchers, payers, planners and educators—to make the changes that will lead to better patient outcomes, better system performance and better professional development (pg 1).” Definitions of health service design also emphasise different organisational stakeholders working together, but with a focus on creativity to imagine and enable innovations in practice or process within organisations, networks or wider service ecosystems [11]. Designing and improving services not only involved internal processes, but also external drivers, such as stakeholders agitating for change through advocacy and activism [12] and the economic and social environment of the health service [13,14] Based on these definitions, it is clear that design and QI in health services is not just the role of service providers or health organisations, but of all stakeholders in health, and is influenced by both internal external factors.

Stakeholder engagement in health service design and QI which includes consumers has been shown to improve patient access and outcomes [15,16,17], and improve patient-centred care [18]. However, there are barriers associated with traditional face-to-face engagement methods, such as insufficient resourcing [16,18,19] and training [19], a lack of representation of people from socially disadvantaged groups [20,21,22], and concerns about the representativeness of small numbers of consumers engaged in activities [17]. Using social media for engagement has been proposed as one way that health services could address some of the limitations or barriers of traditional stakeholder engagement methods [21,23,24,25,26].

The aim of this scoping review was to explore the benefits, risks, barriers and enablers for using social media as a tool for stakeholder engagement in health service design and/or QI. This study complements a previously published paper which used the same scoping review method and described a related but distinct set of findings on the features of social media platforms, models of communication, populations of people, and types of service design, QI or change activities [9]. The key findings from this earlier review [9], with studies included up to 2 February 2019, were that social media was being used for both gathering data to inform service design and QI, and for collaborative decision-making between stakeholders, both of which are common activities within existing QI models [27,28]. The earlier review also found that there are a variety of different engagement approaches available via social media depending on the stakeholder group being engaged, and the communication models and platform features being used. This article updates the search to August 2022 and explores the benefits, risks, barriers and enablers of using social media for service design and QI purposes.

## 2. Methods

This scoping review was conducted using Joanna Briggs Institute (JBI) methodology [29]. The objectives, inclusion criteria and methods of analysis for this review were specified in advance in a protocol [30] and the PRISMA Extension for Scoping Reviews (PRISMA-ScR) checklist [31] is included in Appendix A.

It should be noted that the JBI methodology for scoping reviews was updated in 2020 however this was after the scoping review protocol was published [30] and enacted [9]. In the limitations section of this article we have acknowledged the methodological benefits and limitations of using this review method.

### 2.1. Stakeholder Involvement in This Scoping Review

This review was guided by the involvement of an advisory committee of three healthcare consumers and three healthcare service providers. Members of the advisory had professional and/or lived experience in palliative care, mental health, ICU and critical care, transplantation, chronic disease, consumer engagement and representation, and health communications.

The advisory committee and scoping review were part of a larger doctoral research project. Committee members were invited following purposeful sampling [32] from the researchers’ professional networks. The committee members were invited to be part of the research project because of their roles as service providers or consumer representatives in Australian public hospital settings and because they had experience using social media as part of their role and/or experience in consumer engagement in hospital QI. Invited stakeholders were given detailed participant information and signed a written consent form to confirm their participation.

Advisory committee members contributed to the review by:Deciding upon the research question;Finding articles for screening;Assisting with data analysis, including making suggestions to refine and re-group codes and themes;Reviewing the manuscript; andShaping the content of the discussion section through prioritising the analysis findings based on their own experience of the health system.

Advisory committee member contributions to the review were made through two 90 min meetings held via video conference, and through ongoing email correspondence throughout the review and manuscript-writing process.

The two members of the advisory committee who provided feedback on the manuscript, and answered the questions which informed the discussion section, are included as co-authors on the review (NJ, CL).

### 2.2. Eligibility Criteria

The eligibility criteria outlined below and the subsequent search strategy and methods for the extraction of the results were guided by the Population-Concept-Context structure [29]. The eligibility criteria are summarised in Table 1.

#### 2.2.1. Population

For inclusion in this review, participants in the included studies were:Users or potential users of a health service (i.e., patients, consumer representatives, consumers with an acute or chronic condition, carers, family members, consumer organization member, community members, general public); and/orHealth service providers (health professionals, health service manager/administrator, health policy makers).

No restrictions were placed on the age or gender of study participants.

Studies only involving participants from non-health service settings (e.g., educational institutions, social care services) were excluded from the review.

#### 2.2.2. Concept

The core concept to be examined in this review are the benefits, risks, barriers and enablers of using social media as a tool for health service design and QI. To capture the full range of potential uses of social media in health service design and QI, two broad study types were included. These were:Studies where social media was used as a tool within design or QI activities which were initiated by health organisations, andStudies where social media was used as a tool by stakeholders to influence or advocate for changes to the design or delivery of health services, systems or policy.

A date range limitation of ‘2004–current’ was placed on the search. This date range was chosen because 2004 is considered the beginning of Web 2.0, the era in which the internet shifted from being primarily comprised of static webpages, to instead being made up of sites and applications which allowed for user-generated content creation and distribution [1]. This shift led to the widespread development and use of social media platforms [1].

Studies where social media was used, but there was no intention or objective to design or improve health services, systems or policies were excluded from the review. This included social media being used for disease surveillance, health information dissemination, treatment, peer support, education and research without health service change, design or improvement.

#### 2.2.3. Context

Studies conducted in health service or health policy settings and published in English. Studies from any geographic location, regardless of income status, were included.

Studies conducted in non-healthcare settings (e.g., educational institutions, social care services) were excluded.

#### 2.2.4. Study Design

Original primary research or evaluation articles (any methods) in peer reviewed academic publications, were eligible for inclusion in the review. Sources without original research, and secondary research literature, were excluded from the review.

### 2.3. Search Strategy

On 16 August 2022 Medline OVID, Embase OVID, PsycINFO OVID, and CINAHL EBSCO were searched. All searches were restricted by the date range of January 2004–current (16 August 2022) and published in English only. Example search strategies for Medline OVID and Embase OVID are provided in Appendix A

In addition to the searches above, the reference lists of included studies were screened for potentially eligible studies. Members of the advisory committee also submitted potentially relevant studies for inclusion.

### 2.4. Screening of Studies and Extraction of the Results

L.W. screened the titles and abstracts of retrieved studies, and the screening process was managed in EndNote. L.W. retrieved the full text of potentially eligible studies and assessed them for inclusion, using a screening template developed by L.W., N.H. and S.H. (Appendix A). When an inclusion decision was unclear, two other authors (N.H., S.H.) reviewed the article, completed the screening template, and discussed their assessment with L.W. to reach consensus about inclusion. L.W. performed the data extraction.

### 2.5. Synthesis of Results

The analysis represents the scoping review step of “charting the data” which allows for both a numerical summary of the scope of the literature in terms of quantity of research (Appendix A) and qualitative analysis of the findings of included studies [29,33,34]. A qualitative deductive content analysis was conducted, based on the deductive content analysis method for qualitative systematic reviews described by Finfgeld-Connett [35]. In the first round of coding, data was extracted using a priori themes of ‘benefits’, ‘risks’, ‘barriers’ and ‘enablers’ by L.W. S.H. and N.H. reviewed the coding independently and provided critical input through critical reflexive monitoring and discussions during research meetings. Following this, the advisory committee reviewed the coding and provided feedback on the analysis, with a particular focus on ensuring that the coding aligned with the categories in relation to their knowledge and experience of social media and stakeholder engagement in health services. This feedback was incorporated into the final results.

Comments from the advisory committee in response to the results of the data analysis also informed the content of the discussion section of this article, including how results should be presented and which elements should be emphasised.

### 2.6. Variation in Method from the Published Protocol

In the protocol [30] and the earlier review [9], conference abstracts, secondary research, and grey literature were eligible for inclusion. This decision was made because when the protocol was written, and original search conducted, we believed that there may be a lack of eligible articles in the primary, peer reviewed, research literature alone. However, the 2022 search revealed a large increase in social media research since 2019, with 5456 articles eligible for screening in 2022 compared to 1393 articles in the earlier review [9]. Additionally, the grey literature search in the earlier review returned very few relevant publications [9]. Therefore, a decision was made to focus this review on the primary, peer reviewed, academic literature. This review presents findings from primary, peer reviewed, research literature only.

When an otherwise eligible conference abstract was identified, an additional search was conducted to determine whether a peer reviewed article from the same study had been published. When an otherwise eligible secondary research article was identified, the studies included in the review were individually screened for eligibility.

## 3. Results

### 3.1. Search Results

After screening 5456 titles and abstracts, and 292 full text articles, 61 articles from 59 study settings were included in the scoping review. Figure 1 [36] presents the search results.

All articles were published since 2010, with 23 of the included articles (38%) published since 2020. The majority of studies (90%) were conducted in high income countries. The list of included studies and their key characteristics are shown in Table 2.

### 3.2. Qualitative Content Analysis

The benefits, risks, barriers and enablers for the use of social media as a tool for stakeholder engagement in service design and QI are presented below. Key findings of the analysis are presented in the commentary for each of the a priori themes. A summary of the themes, sub-themes and codes developed through the analysis is presented in Table 3. The frequency of themes, sub-themes and codes reported in the included studies are presented in Appendix A.

### 3.3. Benefits

Using social media for stakeholder engagement in health service design and QI resulted in a broad range of reported benefits. Benefits were grouped into the sub-themes of (1) improves organisational communication, (2) builds relationships, (3) higher quality information and (4) improves organisational culture and reputation.

#### 3.3.1. Improves Organisational Communication

**Improves the efficiency of communication.** The most common benefit relating to organisational communication was that using social media as a tool within health service design and QI activities improves the efficiency of communication [37,40,41,42,45,46,48,49,51,52,54,55,56,57,62,63,66,68,69,71,73,74,77,80,82,84,85,86,87,92,95]. Communication efficiency was improved through increasing the speed of communication [41,42,45,54,57,62,63,69,73,82,92], reducing barriers to face-to-face communication (such as people being in different locations or doing shift work) [37,85,86,92], and increasing the number of people reached [40,42,45,46,48,56,57,71,80,95]. Five studies also found that a response to communication was more likely through social media than other communication methods [49,55,66,71]. Social media was also seen as low cost [42,68,69,73,84], easy to implement [51,52,68] and easy to use [68,69]. One study also found that social media communications also made it easier to redirect queries and link the public to experts, when the organisations involved were unable to respond to the query themselves [84].

**Additional channel for communication.** Social media provides an additional channel for communication [37,48,49,51,54,55,56,58,59,65,66,73,80,84]. As well as simply being another way to reach audiences [48,54,58,80,84], this additional channel is used for gathering information about service quality alongside more traditional methods such as patient surveys [49,55,56,65,66] and supporting or enhancing offline engagement [48,59,73]. Norman et al. [73] observed that this ‘additional channel’ benefit could apply to consumers as well as organisations, finding that the young people involved in their study used their involvement in service design activities as a starting point for building their broader online presence.

#### 3.3.2. Builds Relationships

**Facilitates collaborative relationships.** Social media is beneficial for collaborative relationship building between individuals, groups or organisations [37,40,43,46,48,56,63,66,67,69,70,71,73,77,80,81,82,86,88,89,91,93]. Social media facilitates collaborative relationships in a variety of ways including by fostering conversations and connections between groups who rarely talk [37,43,56,63,67,69,80,82,86,89,91], making mobilisation and collective action easier [46,48,70], strengthening existing relationships [73,77,86], and providing a platform for incidental peer support within QI-focused activities (provider-provider or consumer-consumer support) [40,66,67,71,81,82,86,88,91,93].

**Engages new audiences.** Using social media in health service design and QI also provides opportunities to develop relationships with new audiences [41,42,44,51,52,56,59,62,63,73,80,85,86,88,91]. This includes reaching young people [41,59,73] and community groups known to experience inequities in healthcare access including First Nations people and those living in rural and remote areas [85].

**Improves clinical practice.** The relationship-building capacity of social media was also seen to improve clinical practice [39,69,72,81,91,97]. This occurred through service providers gathering insights into the lived experience of healthcare consumers and improving practice or creating new interventions due to these insights [39,91,97], and by facilitating peer learning between providers [69,72,81].

#### 3.3.3. High Quality Information

**Improves the quality of information gathered or shared.** Social media was seen to improve the quality of information gathered or shared by organisations [46,47,48,49,54,63,65,66,68,69,73,75,76,84,85,86,90,95]. The information gathered from social media relating to QI and service design was viewed as richer and more authentic than information gathered through other methods [46,49,66,68,69,73,85], and social media was perceived to be a good source of additional qualitative information which added context to quantitative performance data [65]. In the context of health policy development, social media was viewed as a good tool for gathering a broad understanding of issues impacting health policy development or change [47,48,56,64,75,84,90,95].

**Facilitates high quality discussions.** Social media was seen as effective for facilitating high quality discussions [39,49,56,67,69,76], through improving two-way communication between service providers and consumers [49,56], creating an environment for generative dialogue [39,67,69], and by creating opportunities for in-depth discussion and evolution of ideas on platforms which allowed for long-duration conversations over weeks and months [76].

#### 3.3.4. Improves Organisational Culture and Reputation

**Facilitates positive organisational culture change.** The use of social media in QI and service design was seen to facilitate positive organisational change [51,52,63,66,69,71,72,73,74,78,81,82,84,92] through increased organisational transparency [51,52,66,69,74,84] and flattening of hierarchies [63,71,81,92]. Using social media to be transparent around organisational processes helped consumers to better understand organisations [51,52], increased trust in organisations [51,52,81,84] and could improve an organisation’s reputation.

### 3.4. Risks and Limitations

The risk and limitations reported in the included studies were grouped into the sub-themes of (1) limited or ineffective engagement, (2) limited evidence of effectiveness, (3) risks of direct harm to individuals and organisations, and (4) challenges to strategic use.

#### 3.4.1. Limited or Ineffective Engagement

**Underutilisation by target audiences.** The most frequently reported risk or limitation theme was that social media is underutilised by target audiences of the health service or QI/design activity [42,45,47,51,52,56,59,60,63,64,65,66,70,74,75,76,78,79,83,84,86,89,90,96,97]. There was a belief the people who used social media were not representative of the stakeholders who would be affected by the change in service design or health policy [42,47,51,52,56,59,60,63,64,66,70,75,76,83,84,86,90,97] and that few people in general used social media [51,52,65,74]. Particular target groups reported to be less likely to use social media were people over 40 years old [79,96,97], people with poor health literacy [96], people of non-Caucasian/Anglo-Saxon ethnicity and culture [89,97] and service providers [74]. Underutilisation was also seen in the ways that people engaged in social media tools. One study [45] observed drop off in social media engagement over time as the activity progressed. Another [78] found that while the social media features of a purpose-built website were valued by participants, in reality they were used far less than other website features (such as information pages or access to tools).

**Quality of discussion or information gathered not sufficient for QI purposes.** The quality of discussion or information on social media could limit its effectiveness for informing QI activities [39,42,56,57,63,65,75,76,80,97]. Information gathered may not be relevant for QI purposes because it might not answer the right question, responses may be too emotionally driven, or people may self-censor because of the public nature of the platform [42,65,76,97]. True co-creation of service design solutions between different stakeholder groups was seen as hard to achieve [39,57], and social media was seen as a place where people with similar views just come and reinforce their ideas and opinions rather than sharing ideas and learning from each other [56,60,63,75,80].

#### 3.4.2. Limited Evidence of Effectiveness

**Unclear evidence of the benefits compared to traditional stakeholder engagement methods** was a limitation to using social media in some of the included studies [42,46,49,50,51,52,54,55,65,66,69,79,80,84,86]. The most common concerns in these studies were about the reliability of the information being gathered. There were mixed findings across four included studies as to the relationship between health service metrics gathered on social media (e.g., sentiment analysis, rankings, reviews) and those gathered through traditional mechanisms (e.g., patient surveys, readmission rates). Two studies found no correlation [49,55], one study found there was a correlation [65] and one study found a correlation in some areas and not others [50]. There are also concerns that some of the artificial intelligence (AI) technology used to gather and classify data through Twitter may not actually be reliable enough to replace human classification [42,51,52,54,55,81,86]. This may be an issue when organisations use data gathered through social media to inform their QI activities, especially organisations dealing with large volumes of data and/or those using social listening methods to gather their data.

#### 3.4.3. Direct Harm to Individuals and Organisations

**Malicious, fake or negative messages and actions** were recognised as a risk [41,42,47,49,51,52,53,64,74,84,86,90,97]. Negative, false or unsavoury messages (that are unable to be removed on some platforms) [42,51,52,53,74,86,90,97] and users misrepresenting themselves [53] were examples of malicious online behaviour from the included studies. Commercial interests posing as regular users, the use of bot accounts, and the deliberate use of misinformation to influence policy debate were also recognised as risks [47,60,64,90]. One study also recognised that—while not intentionally negative—organisations or individuals could lose control of messages, particularly if they went ‘viral’, which could lead to misunderstanding of the message or misinterpretation of the messenger’s intent [41].

**Breaches of privacy and professional behaviour**. Privacy and professional behaviour breaches are a concern for all social media users engaged in service design and QI [38,41,58,74,76,84]. Health services are concerned about maintaining service user confidentiality [38,41,84]. Service providers are concerned about being identifiable online and are also worried about breaching codes of professional behaviour [74]. Consumers are concerned about being identified as a service user [58] and are also reluctant to openly name providers when they’ve had poor experiences, even in a service-improvement context.

#### 3.4.4. Challenges to Strategic Use

**Difficult to use strategically to achieve change.** Social media can be difficult to use in a strategic way to achieve change [42,49,55,63,81,97]. Getting messages to effectively cut through the ‘noise’ of social media and reach the right audiences, and channelling online messages into effective (and often offline) actions, were challenges for organisations and individuals trying to use social media strategically [49,63]. Additionally, people managing social media pages were at risk of being overwhelmed and unable to process data or respond in a meaningful way if social media interaction was very high [81,97].

**Difficult to evaluate.** Social media engagement was seen as difficult to evaluate in five studies [55,65,74,80]. A lack of industry standards across platforms for analytics data [74,80], uncertainty about how online interactions should be analysed [65] and a perception of positive language bias on social media [55] were all seen as risks or limitations to effectively measuring the outcomes of social media use in health service design or QI activities.

### 3.5. Barriers

Barriers to social media use in health service and system design and QI were experienced by all user types. Barriers included specific, observed, barriers which occurred during a study, and expressed concerns based on opinions or beliefs about health service use of social media that were significant enough to stop or limit social media use. The expressed concerns that act as barriers to the use of social media were wide-ranging and were most often expressed by organisations or service providers, rather than health consumers. The barriers identified in the included studies have been grouped into the sub-themes of (1) lack of access to and familiarity with social media, (2) lack of organisational processes and support, (3) concerns about how people behave online, and (4) problems with social media platforms.

#### 3.5.1. Lack of Access to and Familiarity with Social Media

**Lack of resources and access** was the most frequently occurring barrier [39,41,43,51,52,58,69,73,74,78,80,82,86,96,97]. A lack of access to social media platforms (often due to a lack of access to hardware or the internet) was the most common access barrier for all stakeholders [39,51,52,69,73,74,78,80,86,96,97]. Other reported access and resource barriers included lack of time [38,41,43,58,82,97], money [41] and staff [41].

**A lack of skills and confidence in using social media** [39,43,51,52,74,84,97] and a **lack of familiarity with using social media for health or QI purposes** [39,68,72,88] were also observed in the included studies. These barriers were reported by both consumers and provider stakeholders.

#### 3.5.2. Lack of Organisational Processes and Support

**Issues with organisational culture and a lack of executive support** were observed as barriers in some studies [41,43,51,52,61,71,72,84]. The barriers within this theme included a lack of buy-in from the wider organisation [51,52,61,72], organisational cultural issues such as risk aversion or a lack of communication between different groups of people [41,43], governance structures blocking the use of social media within an organisation [41,43], a lack of procedures and guidelines around social media [38,84], and organisational processes which impeded timely responses to social media posts [41,71]. From the consumer point of view, a lack of trust in an organisation makes consumers less likely to want to engage with them through social media [51,52].

**Concerns about implementation and evaluation processes** were expressed as barriers to the use of social media [39,41,50,51,52,65,73]. Concerns were most commonly expressed by service providers, and included worries about added workload and extension of role [39,50,51,52] and doubts about the benefits of web-based tools over usual practice [39,65].

#### 3.5.3. Concerns about How People Behave Online

**Concerns about how to manage messaging, people and interactions in a public forum** were reported as barriers [41,51,52,61,66,68,72,97]. Both logistical concerns and reputational concerns were raised. Logistical concerns were focused on practical management of an online community, such as concerns about managing dominant voices in a discussion [41,66] or finding times for real-time group chats across time zones [68,72]. Reputational concerns were focused on issues such as loss of message control [41], other users disagreeing with the opinions of the individual or organisation [61], legal action [41], managing the discussion of taboo topics [41], and worries about bogus complaints [51,52,97].

**Concerns about the possibility of users breaching privacy or professional codes of behaviour** were expressed as barriers to using social media for health service design and QI activities [41,43,74,91,97].

**Unwillingness to share personal information online** can act as a barrier to consumers using social media for health service design and QI activities [51,52,89,91,97].

#### 3.5.4. Problems with Social Media Platforms

**Rapid changes in the social media environment** meant organisations struggled to maintain a presence across all relevant platforms [68,89].

**Poor usability of platforms**, particularly purpose-built platforms [39] or platforms not being compatible with mobile phones [67], were also identified as barriers.

### 3.6. Enablers

The enablers of social media use in health service or system design and QI were grouped into the sub-themes of (1) facilitating access and use for all stakeholders, (2) making discussions safe, (3) providing high quality content and user incentives, (4) supportive organisational systems and culture, and (5) building a social media community.

#### 3.6.1. Facilitating Access and Use for All Stakeholders

**Making use of social media easier for target audiences.** Making social media easier to use enables its use in health service design and QI activities [38,39,40,43,50,51,52,61,66,67,68,69,73,77,81,86,87,92,96,97]. Social media use can be made easier by providing education and support to users when they’re using unfamiliar platforms or technology [43,50,51,52,61,69,73,81] and by either designing purpose-built platforms specific to user needs [67,73,92,96,97] or choosing commercial platforms with features and functions best suited to the target audience and the style of messaging, campaign or activity [38,66,77,87]. Connecting projects to existing social media groups [40,86] also made it easier for stakeholders to participate in service design or QI activities.

**Organisational systems, processes, resourcing and partnership.** Organisations can enable the use of social media through creating systems, processes, plans and roles related to social media use [38,41,43,51,52,54,65,73,74,81,82,87,93,97]. This includes having guidelines and policies that staff can follow [38,74,97], and having organisational champions and leadership buy-in [38,41,81,82,87]. Organisational strategies such as partnering with similar organisations to run joint social media accounts [51,52], and dedicating time [43,73] and staff [41,43,73,82] to social media activities all support the organisational use of social media.

**Providing multiple ways to engage.** Social media being only part of a suite of engagement strategies was an enabler [39,51,52,60,69,71,73,74,80,85,88]. Providing multiple ways to engage has the potential to reach more people and different audiences, and including a social media option for engagement can increase participation in other modes of engagement (including face-to-face) and vice versa [39,69,73].

**Facilitating access to social media.** Making access to social media easier enabled its use [45,69,73,77,93]. That much social media is already low cost, and available at all hours and on all days was an access enabler for participants [45,73]. Advocating for the widespread availability of software, hardware and internet access [69,73], or even providing hardware to participants in one study [93], were seen as actions that could be taken to ensure access barriers were decreased even further.

#### 3.6.2. Making Discussions Safe

**Making discussions safe.** Safe discussions for all participants was a commonly reported enabler [39,40,49,50,53,55,65,69,71,72,73,81,84,91,93]. Examples of ways that discussions could be made safe were through organisations monitoring and having an active presence on their social media platforms [49,55,65,69,71,72,73,84], discussions being moderated or facilitated [39,65,69,93] and having ground rules for users [39,69]. There was some conflicting information about how to make discussions safe, with one study advocating for transparent and open discussion culture with ‘undisguised’ interactions [72], while others identified the importance of anonymity for some users or users making a new social media accounts just for the activity rather than using their existing professional or personal account [40,69,71,91].

#### 3.6.3. Providing High Quality Content and User Incentives

**Delivering engaging, trustworthy and targeted content.** Quality content, tailored to social media platform formats and communication styles, is a way to enable and encourage use of social media for health service design and QI [39,43,51,52,53,66,68,75,78,80,81,84,86,88,94].

**Users gaining benefits from participation.** All user types were more likely to use social media if they could see benefit from their participation [39,40,41,50,69,73,81,82]. Benefits included incentives for use [39,41,81], sharing of research findings [39,81,82], avoiding more onerous forms of engagement [39], seeing positive results of engagement [53,81,82], and users being appreciated and celebrated for their contributions [40,69,73,81,82].

#### 3.6.4. Building a Social Media Community

**Fostering connections between users within a social media community.** Connections between users enabled the use of social media [38,40,43,68,69,70,72,80,82,88,89,93]. Users were more likely to engage in a social media community if they had an existing friendship or offline connection with at least one other member [38,68,88,89], members of the group had some shared identity [38,40,72,82], and group members could be involved in key groups roles (such as hosting chats, moderation, as champions or advocates, or being willing to post often to lead conversations) [43,69,70,72,80,82,93]. Diversity was important too, because people were more likely to stay in groups where they could relate to the group leadership, and this was particularly important for people who were from cultures or ethnicities different to the majority of group members [89].

**Organisations promoting their use of social media.** Promotion is a way to build a social media community for health service design and QI activities [41,42,51,52,65].

## 4. Discussion

The findings of this review present a comprehensive summary of the scope of existing research into the risks, benefits, barriers and enablers of social media use experienced by stakeholders involved in, or trying to advocate for, the design and improvement of health services, systems and policy.

Benefits of social media use in health service and system design and QI included improvements in organisational communication, culture and reputation, and improvements in relationships and the resulting quality of information from the social media process. The most frequently reported enablers were approaches which made access and use easier and discussions safer. Common negative experiences of use related to the risks of limited engagement and a lack of evidence around social media. In terms of barriers, a lack of access to, and familiarity with, social media, and a lack of organisational support were reported most frequently.

In general, the benefits, risks, barriers and enablers of social media use described in the current literature were similar across user groups, which indicates that solutions to overcome reported risks and barriers are likely to benefit all users. However, further study into the experiences of marginalised population groups is warranted and might reveal different experiences that are not currently known.

Advisory committee members determined the most relevant and important findings from the analysis, based on their experience of stakeholder engagement in health service design and QI and their knowledge around the issues related to social media-based engagement faced by health services and their stakeholders. These findings are emphasised and discussed in detail in the sections below.

### 4.1. The Importance of Engaging New Audiences and Overcoming Underutilisation

Social media may assist in helping health services engage new audiences in the process of QI [41,42,44,51,52,56,59,62,63,73,80,85,86,88,91], but success was mixed with many studies reporting underutilisation by target audiences [42,45,47,51,52,56,59,60,63,64,65,66,70,74,75,76,78,79,83,84,86,89,90,96,97], particularly by people viewed as ‘hard-to-reach’ [98]. Based on their experience as stakeholders involved in health design and QI, our advisory committee believed that the most practical ways to overcome this limitation were educating potential users on how to use social media [43,50,51,52,61,69,73,81] and using social media in addition to other, more traditional methods of engagement (e.g., face to face) rather than relying on social media alone [39,51,52,60,69,71,73,74,80,85,88].

### 4.2. Managing Negative, False or Malicious Messaging

The risk of negative, false or malicious messaging was highlighted as an important finding, even though this risk was only mentioned in thirteen studies [41,42,47,49,51,52,53,64,74,84,86,90,97]. The discrepancy between the relatively small number of included studies which viewed negative, false or malicious messaging as a risk, and the importance placed on this risk by our advisory committee members may reflect issues around delays between conducting and publishing academic research in the context of a rapidly changing digital environment. For example, issues such as the strategic use of misinformation and bots to influence policy debate have only emerged in some of the more recent studies [47,60,64,90]. Other recent social media issues, such as social media algorithms creating highly polarised filter bubbles [99], and selling of user data to third parties [100], were not investigated in the studies included in this review. It is likely that these issues will emerge in future research, particular given the complex advocacy, social support, information- and misinformation-dissemination role that social media has played in the COVID-19 pandemic [101]. Future research should continue to explore emergent risks and barriers to social media use and how they can be managed or overcome.

The advisory committee felt that the strategies of monitoring [49,55,65,69,71,72,73,84], moderation [39,65,69,93] and group rules [39,69] reported within the theme of ‘making discussions safe’ were the most important and relevant ways to manage the risk of negative, false or malicious messaging for stakeholders engaged in health service design or QI activities. Given the perceived importance of these strategies in making social media spaces safer, we would have expected they would be more frequently reported as an enabler for using social media to engage stakeholders in design and QI. This potential under-reporting of moderation, monitoring and group rules may reflect the types of platforms and their access features used in the included studies. Monitoring and moderation are likely easier to undertake and report on in registration-required or closed social media environments, where monitoring and moderation of small online groups is likely done by group members, organisers or paid content moderators [102]. The potential under-reporting of monitoring and moderation may be because the majority of included studies occurred in public social media spaces (see Table 2), where monitoring and moderation may be done through often opaque artificial intelligence mechanisms or anonymous reporting of problematic content [103], and is therefore difficult to report on, or not identified as an enabler in research articles.

Whatever the reason for the potential under-reporting of monitoring, moderation and group rules, the fact that our advisory committee highlighted the importance of these strategies indicates that they may be a key area for future research, in order to better understand the impacts of monitoring and moderation practices on stakeholder engagement in health service and system design and QI.

### 4.3. Building Relationships

Social media can have positive impacts on trust and the quality of relationships between consumers and their providers or health organisations. The potential for social media use to make organisations more transparent [51,52,66,69,74,84], and help consumers better understand how organisations work [51,52], are possible ways to overcome the distrust that some consumers may feel towards health services [104]. Related to this could be the potential of social media to create more collaborative relationships, through both the levelling of hierarchies [63,71,81,92] and facilitating interactions between people and groups who rarely interact [37,43,56,63,67,69,80,82,86,89,91], creating new opportunities for innovation.

However, the findings of the review also demonstrate that this experience of improved relationships is not universal. Some of the included studies identify risks of negative experiences for users through social media messaging or online behaviour [42,51,52,53,74,86,90,97], or just an inability to reach key user groups for relationship building [42,47,51,52,56,59,60,63,64,66,70,75,76,83,84,86,90,97]. These risks and limitations would need to be managed if relationship building through social media were to be successful.

#### Anonymity

There was some discrepancy between consumers wanting some ability to maintain privacy and anonymity in social media spaces [71,91], and providers wanting to restrict the ability for users to be fully anonymous or use pseudonyms [72], although this was only reported in a small number of studies. This discrepancy may be reflective of the type of activities being conducted and the participants involved. For example, both of the studies which advocated for anonymity in social media spaces examined the experiences of consumers who shared experiences of care to try and influence change at the health service [71] or public health policy [91] levels. Moorley et al. (2014) found value in undisguised interactions between members of a health professional group using social media for networking and seeking professional advice [72]. Discrepant findings are expected given the range of audiences, populations and issues reviewed and the complexity of modes of social media use [9]. Future efforts by services could be informed by models and examples that are relevant to their needs and aims [9] and by planning social media use in health service design and QI activities in partnership with advisory groups of clinical and consumer stakeholders [105].

### 4.4. What This Review Adds to the Literature

This is the first scoping review to our knowledge to explore risks, benefits, barriers and enablers for using social media as a tool within health service design and QI activities. Much of the existing literature which examines social media use in health either discusses theoretical models of use in service design or QI activities [23,106,107,108], or explores other aspects of health-related social media use, such the impacts of social media use on patient outcomes [6,7,8]. This review adds to the broader health-related social media use literature by bringing together and analysing current research findings to inform practice, policy, and research.

The findings of this review challenges previous theoretical literature on the use of social media in health service design and QI. Much of the theoretical literature views social media as a way to overcome some of the limitations or barriers of traditional face-to-face models of consumer engagement [21,23,24,25,26]. However, many of the identified barriers to face-to-face consumer engagement—such as insufficient time, staff and financial resources [16,18,19], a lack of experience and training [19], and a lack of consumers from marginalised groups being engaged [20,21,22]—were also identified as barriers or limitations to the use of social media for engagement. It may be that social media use by itself is not a solution to overcoming barriers to consumer engagement in health, but that social media should be considered as part of a larger solution to the problem of how the involvement of service users in health service design and QI activities is made more effective, meaningful, and attractive to consumers and organisations alike.

Additionally, the findings of the review reinforce what is known around the use of social media in health more broadly, such as the benefits of improved data gathering and sharing, improved relationships and support, and concerns about privacy, usability and misinformation, which are already reported in research on social media-based health peer support and health service delivery activities [6,7,8].

### 4.5. Gaps in the Literature

Few included studies reported monitoring and moderation as an enabler. It may be under-reported because moderation is simply seen as usual business by the organisations and groups that manage social media, and so the use of moderators was not explicitly stated. Given that moderation has the potential to mitigate some of the risks and barriers of social media use (e.g., malicious and fake messaging, breaches in privacy), but may also create or reinforce power structures in online spaces which can affect the quality of engagement [109], more attention to researching the role and impacts of moderation in health-related social media spaces is warranted.

### 4.6. Limitations

There were mixed findings on some key issues. For example, some studies found social media use improved the richness and authenticity of the information gathered for QI and service design organisations [46,47,48,49,54,63,65,66,68,69,73,75,76,84,85,86,90,95], while others found that the quality of the information was not always sufficient for QI purposes [39,42,56,57,63,65,75,76,80,97]. These contradictory findings may reflect the complex nature of social media-based stakeholder engagement, or the breadth of communication models, social media platforms, community engagement practices, groups of people that were examined and research methods adopted in the included studies. Because this is a scoping review and quality assessment of included studies was not required [29,34], the findings cannot determine which benefits, risks, barriers and enablers are most important for the effective implementation of social media-based engagement. Future research into specific models to determine the importance of different implementation factors, and/or qualitative evidence synthesis which includes quality assessment of included studies, may help researchers and implementers determine best practice in social media-based stakeholder engagement in health service design and QI.

The objectives and inclusion criteria for this review meant that the views, opinions and experiences of people who are not current social media users, but who are involved (or who could be involved) in health service design or QI through other engagement methods, were most likely not captured in the included studies. By not including this group, key experiences may have been missed, particularly ones relevant to increasing social media uptake by non- or infrequent-user groups, or understanding experiences that may lead to stakeholders not engaging through social media at all. Additionally, the included studies largely reported on the process of social media-based engagement methods, rather than outcomes of social media-based engagement, or comparisons between social media engagement methods and more typical engagement methods. This means that the findings of the included studies may reflect some pro-social media bias, as they were conducted in organisations or with stakeholders who had already decided to use social media as an engagement tool.

This review had some methodological limitations. The review protocol was published and reviews guided by an earlier version of the JBI Reviewer’s Manual [29] using a single reviewer for title/abstract and full-text screening. Rigour was enhanced by regular research meetings to monitor the screening process, discussion of any articles that were not clearly included/excluded, and the researchers who conducted the screening and monitoring process were all very experienced in conducting systematic, scoping and Cochrane reviews. One benefit of this approach was the single reviewer process increased the speed of review completion [110,111], however it might have led to some relevant studies being excluded [110,111,112].

Finally, using a deductive content analysis method with a priori themes as a coding framework, may have led us to overlooking opportunities to expand or refute existing theories [35]. To minimise this potential limitation of our analysis method, we ensured that any important data that did not align with the a priori categories was coded separately and discussed by the research team to see if a new category, or rearrangement of sub-themes or codes, was warranted. This met the deductive content analysis aim of “alterability” which allows researchers to test, adapt, expand and improve upon their existing framework as the analysis progressed [35].

### 4.7. Implications for Practice

This review adds to the growing empirical literature on the use of social media for consumer engagement in health service design and QI, as well as potentially for broader engagement purposes. Health services and stakeholders involved in design or QI activities could use the findings to inform their planning. For example, the findings relating to benefits may provide support for social media as an engagement approach, while the enablers give practical strategies for implementing social media as a stakeholder engagement tool. The risks/limitations and barriers findings may help planners understand where they need to provide support to help people engage or may meet resistance to social media engagement approaches.

### 4.8. Implications for Research

This review has uncovered a number of potential areas for future research. The highly dynamic nature of social media makes it likely that new issues and experiences around the use of social media as an engagement tool in health service design and QI will emerge in the future. Future research could examine innovations in social media-based engagement and how new risks and barriers to use could be overcome. In particular, the increase in health services being provided digitally and/or remotely during the COVID-19 pandemic [113] may lead to new innovations in social media-based engagement.

While studies in this review described benefits around involving new people in engagement activities [41,42,44,51,52,56,59,62,63,73,80,85,86,88,91], there was little specific reporting of the benefits, risks, barriers and enablers experienced by groups of people known to be at risk of inequities in health care [98]. Given that one of the theoretical benefits of using social media for stakeholder engagement is the potential to increase inclusion of marginalised population groups in the design and improvement of health services [21], further study into the experiences of populations at risk of unequal access to healthcare is warranted.

This review has also identified potential under-reporting of monitoring and moderation as an enabler for stakeholder engagement through social media. Further research into the role and impacts of monitoring and moderation of health-related social media spaces may lead to a better understanding of the role social media moderators play in facilitating stakeholder engagement in health service design and QI.

## 5. Conclusions

This scoping review examined the risks, benefits, barriers and enablers when using social media as a tool to engage a range of stakeholders in health service design or QI activities. The benefits, risks, barriers and enablers described in the included studies highlighted the complex experiences of individuals and health service engaging in service design or improvement through social media. Social media may improve organisational communication and help individuals build relationships which can enhance health service design and QI activities, but can also expose health services and individuals to reputational risks, trolling, harassment and breaches in privacy. These risks can be managed through strategies such as monitoring and moderation of social media spaces and establishing organisational policies and processes around social media. However, without executive support and good communication between implementing teams, concerns about these risks can become barriers to using social media as stakeholder engagement tool, particularly in health services with an existing culture of risk aversion. In addition to these potential barriers caused by organizational culture, health services using social media in their suite of engagement strategies need to consider whether there are any accessibility and usability barriers in any of their stakeholder groups, including access to internet, access to computers and other devices, and the usability or accessibility of social media platforms.

Having a good understanding of the complexity of using social media as a stakeholder engagement tool in the design and improvement of health services may assist planners and implementers to be more aware of, and more able to overcome, some of the known barriers, risks and limitations. It is also important for implementers to understand that there may be experiences that are not yet captured, due to the dynamic social media environment and lack of data on the experience of marginalised groups.

## Figures and Tables

**Figure 1 ijerph-19-14851-f001:**
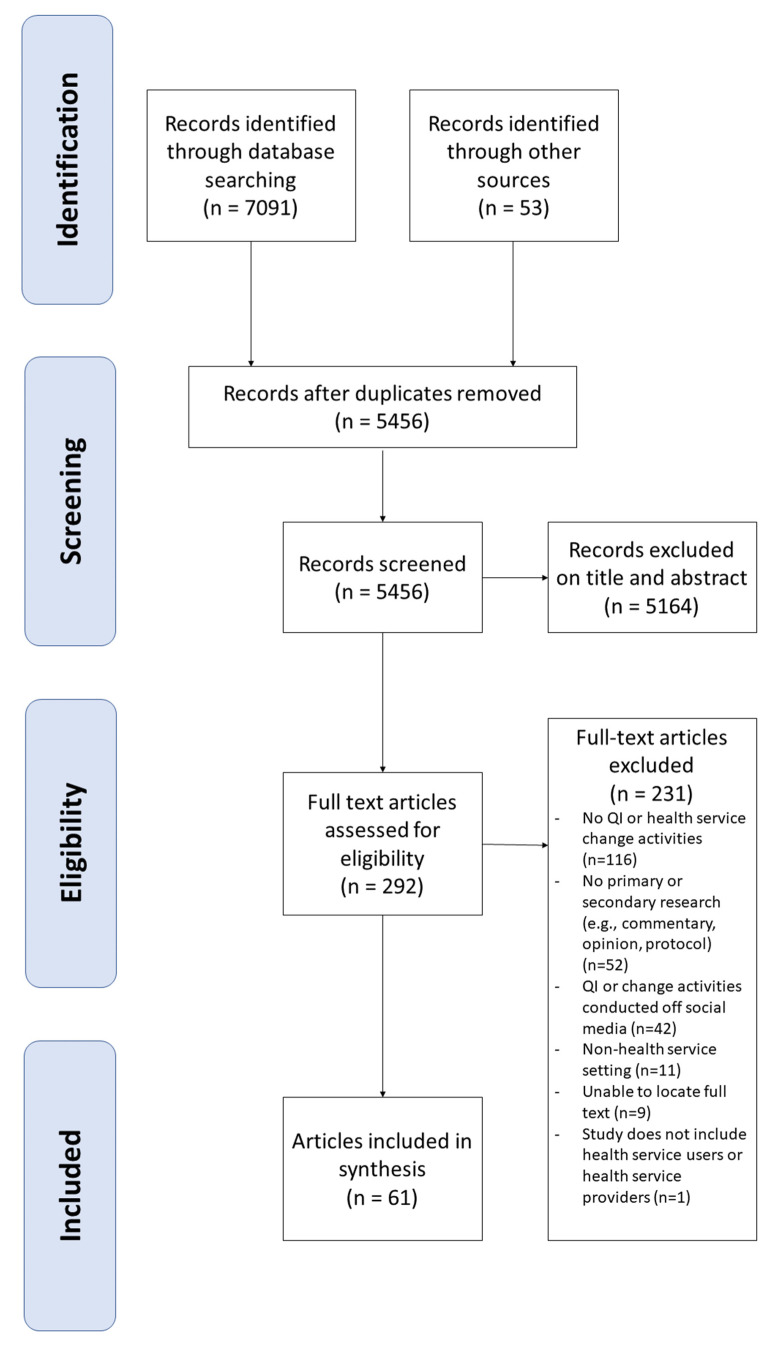
PRISMA Flow diagram. Source from [36].

**Table 1 ijerph-19-14851-t001:** Scoping review eligibility criteria.

Population	Users or potential users of a health service (i.e., patients, consumer representatives, consumers with an acute or chronic condition, carers, family members, consumer organisation member, community members, public); and/or Health service providers (health professionals, health service manager/administrator, health policy makers). Any age, any gender.
Concept	Studies where social media was used as a tool within workforce-based health service design or QI activities; and Studies where social media was used as a tool for communications by stakeholders to influence or advocate for changes to health service design or delivery. Studies published since 2004.
Context	Healthcare or health service setting (hospitals, health services, aged care, community health, primary health, health-specific non-government organisations). Health policy setting (government health departments). Studies published in English. Any geographic location.

**Table 2 ijerph-19-14851-t002:** Key characteristics of included studies.

First Author (Year)	Study Aim	Social Media Platform(s)	Study Design + Participants	Region or Country
Al Fannah (2022) [37]	To share the experience of QI teams during the COVID-19 pandemic	WhatsApp	Mixed; case study Multiple Agile Teams of 6–12 members each	Oman
Ali (2022) [38]	To evaluate a QI project to implement an intervention to assist so doctors to source the one ECG machine on a ward.	WhatsApp	Mixed; case study 12 junior doctors	England, UK
Amann (2017) [39]	To explore the views and experiences of online disability community managers in relation to knowledge co-creation	Forums (unspecified)—registration required (two communities); no registration required (two communities)	Qualitative; semi-structured interviews 9 interview participants	Unspecified
Bahk (2015) [40]	To assess the potential for participatory epidemiology in post-marketing medical device surveillance through engagement with an online patient community	Facebook group, MedWatcher website	Mixed; analysis of posts on Facebook group and MedWatcher website 17,850 Facebook group members; 1354 user submissions on MedWatcher website	USA
Booth (2017) [41]	To explore the experiences and evidence around the use of social media in Ontario public health	Unspecified social media	Qualitative; group deliberation activities and nominal group technique activity 50 participants in the group activities	Canada
Bridge (2021) [42]	To explore the #SugarTax debate on Twitter during the implementation of the UK Soft Drink Industry Levy.	Twitter—public posts and feeds	Mixed methods; social network analysis and thematic analysis of tweets 5366 tweets	UK
Deerhake (2020) [43]	To evaluate the impact a social media–based community of practice had on a post-merger intensive care unit work environment.	Facebook—closed group	Mixed methods; pre-post implementation survey, social media analytics and content analysis 14 participants in the community of practice	USA
Dennis (2022) [44]	To examine attitudes towards a twice-weekly mass asymptomatic COVID-19 testing policy	Twitter, Facebook, comments sections from national newspapers.	Qualitative analysis of social media posts. 5783 comments: 485 comments from Twitter; 3776 comments from Facebook; 1522 comments from newspaper articles	England, UK
De Sousa (2018) [45]	To evaluate a social media-based QI project around the knowledge of best practices in hypoglycaemia management for nurses	Facebook—Public organisational page Instagram—Public organisational page	Quantitative; social media analytics and knowledge survey 101 pre-intervention survey participants 60 post-intervention survey participants	Canada
Doan (2022) [46]	To examine Twitter posts to investigate how users engaged in activism in response to the passage of Georgia’s abortion ban.	Twitter—public posts and feeds	Qualitative analysis of 583 tweets	USA
Dobbs (2022) [47]	To explore Twitter discussions about the federal Tobacco 21 law in the USA in the period leading up to the law being enacted	Twitter—public posts and feeds	Qualitative analysis of 1300 tweets	USA
Gonzalez-Aguero (2022) [48]	To explore a social movement which called for the inclusion of the insulin pump into a universal health cover plan	Various	Qualitative; Semi-structured interviews with nine people involved in the campaign	Chile
Greaves (2014) [49]	To examine and compare tweets sent to hospitals with established measures of quality of care and patient experience	Twitter—public posts and feeds	Mixed methods; volume and frequency of total tweets in study period, and content analysis Content analysis of 1000 tweets	UK
Harris (2014) [50]	To evaluate the implementation of a online food poisoning surveillance, reporting and management system in Chicago	Twitter—public posts and feeds	Quantitative; Twitter and website analytics, outcomes of food safety inspections of restaurants 270 tweets, 193 website complaints, 133 restaurant inspections	USA
Harris (2017) [51], Harris (2018) [52]	To evaluate the implementation of a online food poisoning surveillance, reporting and management system in St Louis	Twitter—public posts and feeds	Mixed methods; Twitter and website analytics, outcomes of food safety inspections of restaurants (2017); interviews with stakeholders (2018) 193 tweets, 7 interview participants	USA
Harris and Moreland-Russell (2014) [53]	To examine the social media response to the Chicago Department of Health e-cigarette campaign	Twitter—public posts and feeds	Mixed methods; Twitter analytics and content analysis of tweets 683 tweets	USA
Hatchard (2019) [54]	To examine how the volume, sentiment and purpose of tweets about standardised packaging of tobacco changed following the announcement of a parliamentary vote on the policy	Twitter—public posts and feeds	Mixed methods; content analysis and descriptive statistics with comparison between findings at two time-points 1038 tweets	UK
Hawkins (2016) [55]	To assess the use of Twitter posts related to patient experience and sentiment as an additional source of data for measuring perceived quality of care in hospitals	Twitter—public posts and feeds	Mixed methods; machine learning classification of identified tweets, calculation of sentiment, surveys of representatives from included hospitals 404,065 tweets, 147 survey participants	USA
Hays (2015) [56]	To identify and describe the range of opinions expressed about care.data on Twitter for the period during which a delay to the project was announced, and provide insight into the strengths and flaws of the project.	Twitter—public posts and feeds	Qualitative analysis of 3537 tweets from 904 contributors	UK
He (2020) [57]	To report on the outcomes of #GetMePPE, which used digital tools (including social media) to advocate for the provision of PPE to hospital workers during the COVID-19 pandemic	Twitter—public posts and feeds	Mixed methods; case study >1800 hospitals and PPE suppliers entered in database and >10,000 signatures on petition associated with campaign	USA
Hedge (2011) [58]	To explore the effectiveness and challenges of a social media-based youth sexual health service evaluation project	Facebook; MySpace; Bebo; Hi5—all public organisational pages	Quantitative; survey 78 survey participants	UK
Hildebrand (2013) [59]	To describe the participatory methods used to involve young people in the UNAIDS strategy planning.	Facebook; Blog; RenRen; Vkontake; forums on purpose-built website—all public organisational pages or open forums	Qualitative; thematic analysis of online and offline forum discussions 3479 participants across all activities	Regional forums for Africa, Latin America, Brazil, Asia Pacific, Eastern Europe, Central Asia, Middle East, North America, Caribbean, Central Europe, China.
Jun (2022) [60]	To identify the Twitter users who engaged in conversations about the Food and Drug Administration’s Modified Risk Tobacco Product Authorisation of the IQOS tobacco heating system related conversations and characterise their tweets	Twitter—public posts and feeds	Mixed; qualitative analysis of tweets, descriptive statistics and use of machine learning 548 tweets	USA
Kearns (2021) [61]	To assess how California dentists use social media to engage in discussions about sugar restriction policies	Unspecified social media	Quantitative; survey 624 survey participants	USA
Khasnavis (2017) [62]	To assess public response to lung cancer guidelines through Twitter	Twitter—public posts and feeds	Mixed methods; Twitter analytics, content and sentiment analysis 172 included tweets	USA
King (2013) [63]	To examine how Twitter users influenced or informed opinions on the Health and Social Care Bill in the UK	Twitter—public posts and feeds	Mixed methods; Twitter analytics, sentiment analysis 120,180 included tweets, 200 tweets included in sentiment analysis	UK
Kirkpatrick (2021) [64]	To document the Twitter conversation pertaining to the proposal or implementation of policies restricting the availability of flavoured e-cigarette products	Twitter—public posts and feeds	Qualitative analysis of 2500 tweets from 536 users	USA
Kleefstra (2016) [65]	To explore how review on hospital rating sites can inform the supervision of hospitals by healthcare inspectors	Hospital rate and review site—public site	Qualitative; semi-structured interviews 10 interview participants	The Netherlands
Lagu (2016) [66]	To understand how Facebook could be used to engage patients in hospital quality improvement	Facebook—public organisational page	Qualitative; thematic coding of Facebook posts 47 posts from 37 respondents	USA
Lara (2017) [67]	To evaluate the implementation of two virtual communities of practice	Purpose-built platform—registration required	Mixed methods; site analytics, survey, case study 66 participants in pilot phase, 181 participants post-launch	Spain
Levine (2011) [68]	To describe the process of using social media to involve young people in the development of an internet-based intervention	MySpace—registration required forum and chat functions	Qualitative; online focus groups 36 focus group participants	USA
Li (2019) [69]	To describe the process of using social media to improve the integration of HIV services for people who inject drugs through the development of a virtual network of antiretroviral therapy and methadone maintenance treatment providers	Facebook—private groups	Qualitative; content analysis of posts 88 treatment provider participants	Vietnam
Litchman (2020) [70]	To examine the Do-It-Yourself patient-led innovation movement through analysis of tweets from #WeAreNotWaiting and #OpenAPS tweets	Twitter—public posts and feeds	Mixed methods; analysis of 46,578 tweets from 7886 participants	International (Tweets for 142 countries)
Mazanderani (2021) [71]	To explore the views of healthcare service users about the relationship between online feedback and care improvement	Unspecified social media	Qualitative; semi-structured interviews 37 interview participants	UK
Moorley (2014) [72]	To evaluate the use of Twitter to create an online nursing community	Twitter—public posts and feeds	Quantitative; Twitter analytics 7000 Twitter followers	UK
Norman (2012) [73]	To present the ways in which young people have been engaged in research and evaluation through the use of social media and digital tools	Facebook; Twitter; Ning; Flickr; purpose-built platform—mix of restricted access groups and public pages	Mixed methods; two case studies Total number of participants not stated	Canada
O’Connor (2017) [74]	To examine social media as a tool to engage nurses in priority setting and policy communication and development	Twitter—public posts and feeds	Mixed methods; Twitter analytics and thematic analysis 64 participants, 444 tweets	Scotland, UK
Olszowski (2022) [75]	To conduct analysis of Twitter discussions around the introduction of mandatory vaccinations for COVID-19 in Poland	Twitter—public posts and feeds	Mixed methods analysis of 71,908 tweets from 21,779 users	Poland
Owolabi (2014) [76]	To explore stakeholder views around the integration of Traditional Birth Attendants into formal health systems	Email moderated forum—registration required	Qualitative; thematic analysis 193 forum participants, 658 messages	Africa, North and South America, Europe
Pisano (2014) [77]	To use social media to increase medical residents’ awareness of antimicrobial stewardship tool and care pathways	Facebook; Twitter—public organisational pages and feeds	Quantitative; pre- and post-intervention survey 39 survey participants	USA
Porterfield (2017) [78]	To evaluate an online QI communication platform for public health professionals	Purpose-built platform—registration required	Mixed methods; survey, interviews and platform analytics 462 survey participants, 21 interview participants	USA
Ramirez (2015) [79]	To evaluate the impact of an online network on health policy advocacy	Facebook; Twitter; Blog; YouTube—all public organisational pages	Quantitative; survey 148 survey participants	USA
Ramirez (2021) [80]	To understand how tweetchats are used to promote dissemination of culturally relevant information on social determinants of health	Twitter—public posts and feeds	Mixed methods; social media and website analytics, network analysis 187 chats with 24,609 users, 177,466 tweets	USA
Rasheed (2021) [81]; Rasheed (2022) [82]	To describe the design, implementation, and evaluation of a social media-based communication and leadership development strategy in a tertiary care hospital in Pakistan	Facebook—closed group	Mixed methods; case study and evaluation (social media analytics and content analysis of posts) 9085 Facebook posts, 625 members	Pakistan
Robin (2022) [83]	To use social media sources to evaluate the acceptance of COVID-19 asymptomatic testing policy in Liverpool	Twitter, Facebook, online newspaper comment section—public posts and feeds	Qualitative analysis of 1096 posts and comments (219 newspaper comments, 472 Facebook comments, and 405 tweets)	England
Shan (2015) [84]	To examine the use and impact of social media on communication between consumers and public food safety and nutrition organisations.	Various—including Facebook, Twitter, Youtube; public posts and feeds	Qualitative; semi-structured interviews with 16 professionals from five national food safety and nutrition organisations	UK and Ireland
Shields (2010) [85]	To describe the use of social media to conduct a healthcare priority setting activity	Facebook; Blog, YouTube, Choicebook; forum (unspecified)—access features undescribed	Mixed methods; platform analytics, thematic analysis of written contributions >800 people	Canada
Shimkhada (2021) [86]	To use Twitter to set priorities and policy recommendations to improve metastatic breast cancer care	Twitter—public posts and feeds	Mixed methods; social media analytics and content analysis 288 Tweets from 42 users	USA
Sivananthan (2021) [87]	To evaluate a communication network between junior doctors and senior leadership established during the COVID-19 pandemic	WhatsApp—access restricted to invited users	Mixed methods; case study and evaluation (social media analytics and content analysis of posts) 780 members of the WhatsApp group	UK
Smith-Frigerio (2020) [88]	To explore how two grassroots mental health groups incorporate advocacy strategies into their social media messages, and how these messages are received by audiences	Facebook, Twitter—access features undescribed	Mixed; content analysis of social media posts and semi-structured interviews 20 interview participants, 200 social media posts	USA
Sperber (2016) [89]	To identify the characteristics of an online health community that may inform and support patient-centred healthcare	Twitter—public posts and feeds	Qualitative; ethnography of Twitter content and semi-structured interviews 22 interview participants	Global, most activity from USA and Europe
Sun (2021) [90]	To explore the Twitter reactions to Australian government plans to prohibit the personal importation of nicotine vaping products	Twitter—public posts and feeds	Mixed; qualitative coding and descriptive statistics from 1168 tweets	Australia
Sundstrom (2016) [91]	To examine the role that sharing health experiences can play in an online advocacy community	Tumblr—public posts	Qualitative; content analysis of blog posts 1110 posts included in analysis	USA
Timimi (2015) [92]	To explore the use of a social media platform to foster cultural change and improve patient experience	Purpose-built platform—registration required	Mixed methods; surveys of knowledge and awareness, platform analytics, thematic analysis of forum posts 254 participants in pilot social media platform	USA
Vasilica (2020) [93]	To determine whether a social media hub for people with chronic kidney disease, which had been co-designed with patients, met patients’ information needs and improved health and social outcomes.	Privately developed platform, Facebook; Twitter—mix of registration required platform, closed Facebook groups, and public Twitter posts and feeds	Mixed methods; co-design and longitudinal evaluation (which included observations on use of the platform, survey and interviews) 15 users were involved in co-design activity; 50 at launch event, 14 in longitudinal study	UK
Waddell (2019) [94]	To describe and analyse social media content from USA nursing organisations in the lead-up to the 2016 presidential election	Facebook; Twitter—public organisational pages and feeds	Qualitative; content analysis of social media posts 2137 posts analysed	USA
Weiler (2013) [95]	To describe the use of a social media strategy to increase feedback to a local draft health and wellbeing strategy	Blog, online poll, Twitter	Mixed methods; case study of 126 poll votes and 3373 engagement responses to the strategy	England
Wu (2019) [96]	To evaluate the acceptability of, and gather feedback on, a gay-friendly health services platform	WeChat	Mixed methods; survey and focus groups with 34 participants	China
Zakkar (2022) [97]	To explore healthcare provider and administrator perspectives on patient stories on social media	Various platforms which allow sharing of patient stories	Qualitative; semi-structured interviews with 21 healthcare providers and administrators	Canada

**Table 3 ijerph-19-14851-t003:** Experience of social media use—themes, subthemes and codes.

A Priori Themes	Sub-Themes	Codes
Benefits	Improves organisational communication	Improves the efficiency of communication Provides an additional channel for communication
	Build relationships	Facilitates collaborative relationships Engages new audiences Improves clinical practice
	Higher quality information	Improves the quality of information gathered or shared Facilitates high quality discussions
	Improves organisational culture and reputation	Facilitates positive organisational culture change
Risks/Limitations	Limited or ineffective engagement	Underutilisation by target audiences Quality of discussion/information gathered not sufficient for QI purposes
	Limited evidence of effectiveness	Unclear evidence of the benefits compared to traditional stakeholder engagement methods
	Direct harm to individuals and organisations	Malicious, fake or negative messages and actions Breaches of privacy and professional behaviour
	Challenges to strategic use	Difficult to use strategically to achieve change Difficult to evaluate
Barriers	Lack of access to and familiarity with social media	Lack of resources and access Lack of skills and confidence in using social media Lack of familiarity with using social media for health or QI purposes
	Lack of organisational processes and support	Issues with organisational culture and lack of executive support Concerns about implementation and evaluation processes
	Concerns about how people behave online	Concerns about managing messages, people and interactions in a public forum Concerns about privacy and professional behaviour Unwillingness to share personal information online
	Problems with social media platforms	Rapid changes in the social media environment Poor platform usability
Enablers	Facilitating access and use for all stakeholders	Making use of social media easier for target audiences Organisational systems, processes, resourcing and partnership Providing multiple ways to engage Facilitating access to social media
	Making discussions safe	Making discussions safe
	Providing high quality content and incentives	Delivering engaging, trustworthy and targeted content Users gaining benefits from participation
	Building a social media community	Fostering connections between users in a community Organisations promoting their use of social media

## Data Availability

Not applicable.

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
