# Peer review of "The Experiences of Stakeholders Using Social Media as a Tool for Health Service Design and Quality Improvement: A Scoping Review"

_ijerph, 2022, doi:10.3390/ijerph192214851_

Round 1

Reviewer 1 Report

This is an excellent scoping review of the existing state of the art literature. The study design and execution is appropriate and supports the goals of the work. All steps and aspects are comprehensively motivated and explained by the authors. Resulting in a clear set of findings and results that are very well categorized. The only very minor question I ask is, why was the period of collecting the sample set from 2004, is there any specific reason? 

Author Response

Thank you for your review. We have prepared on response to all the recommended revisions from the peer reviewers and the editorial team, which is attached. 

Reviewer 2 Report

The paper deals with an important topic, it is well-structured and methodologically sound.

I would only suggest adding a paragraph in the conclusion part and specifying the implications and suggestions specifically for health services, not for organizations in general. 

The anti plagiarism program Plagscan shows a 10% similarity with other articles, please check for similarities and decrease the percentage. 

Author Response

(The authors gave the same response as above.)
